# The Impacts of Viscoelastic Behavior on Electrokinetic Energy Conversion for Jeffreys Fluid in Microtubes

**DOI:** 10.3390/nano12193355

**Published:** 2022-09-26

**Authors:** Na Li, Guangpu Zhao, Xue Gao, Ying Zhang, Yongjun Jian

**Affiliations:** 1College of Sciences, Inner Mongolia University of Technology, Hohhot 010051, China; 2School of Mathematical Science, Inner Mongolia University, Hohhot 010021, China

**Keywords:** streaming potential, periodic pressure, electrokinetic energy conversion (EKEC) efficiency, Jeffreys fluid, nanofluid

## Abstract

In this paper, the electrokinetic energy conversion (EKEC) efficiency, streaming potential of viscoelastic fluids in microtubes under an external transversal magnetic field, and an axial pressure gradient are investigated. The Jeffreys fluid is applied to model the viscoelastic fluid, and the analytic solution of velocity field is obtained using the Green’s function method. The influence of different dimensionless parameters, for instance, the Deborah numbers *De* and *De*^*^, which are related to the relaxation time and retardation time, respectively; the dimensionless electro-kinetic width *K*; the dimensionless frequency *ω*; the volume fraction of the nanoparticles *φ* and the dimensionless Hartmann number *Ha*; and three different imposed axial periodic pressure gradients (cosine, triangular, and square) on fluid dynamics are discussed. The physical quantities are graphically described, and the influence of different parameters on the EKEC is analyzed. The results indicate that *De* promotes the streaming potential and EKEC efficiency of the microtube, while *De*^*^ inhibits them.

## 1. Introduction

Microfluidics refer to the flow of fluid in microscale pipes, which has obtained extensive notice on account of its extensive application in biomedical, chemical, and biological fields [1,2,3,4,5,6]. It is known that, when the solid surface electrolyte solutions of the microchannel come into contact, the solid surface is generally negatively charged. The charged solid surface attracts cations in the electrolyte solution while repelling anions in the solution, thereby forming an electric double layer (EDL) at the solid–liquid interface with a non-zero net charge. The EDL is composed of a Stern layer and a diffuse layer, and the ion density distribution in the diffusion layer obeys the Boltzmann distribution in a quasi-steady state. At present, there are numerous driving and controlling technologies for microfluidics that embrace various principles. Microfluidic flows can be effectively driven by utilizing pressure gradients, surface tension, external electric fields, external magnetic fields, and appropriate combinations thereof [7,8,9,10,11]. The introduction of the outside electric fields to actuate fluid movement generates the fundamental electrodynamic phenomenon called electro-osmotic flow (EOF). On this point, EOF is extensively applied as the driving power for fluid movement and transportation due to its numerous operational advantages [12,13,14]. EOF can also be generated by the motion of the electrolyte solution actuated by a pressure gradient across the microchannel without applied outside electric fields. A current in the same direction as the flow will be formed in the EDL, which is called the flow current. With the flow of the electrolyte solution, the net charge accumulates downstream of the micro-channel, making the downstream potential of the micro-channel higher than the upstream, so the electric fields inverse to the initial fluxion orientation will be formed, called the streaming potential. Because the existence of these induced electric fields, the ions in the solution are obtained for the inverted electric field force, thereby forming a current in the contrary orientation to the initial fluxion, which is electrical conduction. Electrical conductivity provides a simple and effective process for converting mechanical energy generated by stress and chemical energy induced by EDL into electrical conductivity to generate electricity. This process is called electro-kinetic energy conversion (EKEC), which provides new avenues and methods for people to explore new energy and renewable energy. A lot of researchers have studied EKEC efficiency and streaming potential of fluids in microtubes [15,16,17,18,19,20,21,22].

As nanomaterials and nanotechnology have developed rapidly, a fresh fluid has emerged, which is called nanofluid. Nanofluid is a comparatively stable suspension with high thermal conductivity formed by adding a certain proportion of solid particles with a particle size of less than 100 nm in the base liquid. Compared with traditional fluids, nanofluids have more excellent characteristics and have broad application prospects in the chemical industry, energy, and other fields, and are an important research content of many scholars [23,24,25,26,27,28]. Habibishandiz et al. [29] investigated the mixed convective heat transfer of MHD nanofluids containing oxygenated microorganisms in vertical annular porous tubes, and the results demonstrate that the presence of microorganisms leads to attenuation of heat transfer, while an increase in Hartmann number leads to an improvement in the average Nusselt number. Chen et al. [30] studied the theoretical model for reasonably predicting the thermal radiation characteristics of nanofluids with nanoparticle aggregation.

Microfluidic equipment is frequently applied to study colloidal suspensions, biological fluid, and polymer solutions, which are predominantly solutions of long-chain molecules, making the fluids exhibit many properties of viscoelastic fluids. Viscoelastic fluids exhibit both viscous and elastic properties and retain memory of their flow history, which is a non-Newtonian fluid. High molecular polymer fluids include molten plastics, motor oils with polymer additives, and biological fluids such as egg whites, blood, etc. are viscoelastic fluids. The Jeffreys fluid model is a straightforward viscoelastic model. Due to its wide application in medicine, biology, industry, and other fields, the study of viscoelastic fluids has attracted more and more scholars’ attention [31,32,33,34,35,36,37,38]. The Jeffreys model (in other terminology, the Oldroyd-B model) is well known. In previous studies, numerous researchers have studied the fluid flow problem in steady state. The Jeffreys (Oldroyd-B) model with an objective derivative has been studied by various scholars, such as Jaumann derivative, upper convective derivative, lower convective derivative, etc. [39,40,41,42,43]. A three-dimensional steady flow mathematical model of an incompressible Oldroyd-type viscoelastic fluid in a bounded region was studied by Evgenii [44]. Angiolo et al. [45] studied three-dimensional nonlinear viscoelastic models including stress relaxation and creep/recovery phenomena. Kaushik et al. [46] combined the elasticity and viscosity of viscoelastic fluids to study the initiated electroosmotic flow of viscoelastic fluids in a rotating rectangular microfluidic channel described by the Oldroyd-B model. However, the linear Jefferys model with respect to time derivatives has also yielded countless results [47,48,49]. Mederos et al. [50] performed a theoretical analysis of the mass transport rate of lengthy microcapillaries with reactive walls connecting two reservoirs with different concentrations of electrically neutral solutes caused by oscillating electroosmotic flow of Jeffreys fluids. Koner et al. [51] investigated the effect of ion partitioning on solute transport phenomena in fractional Jeffreys fluids. Liu et al. [52] studied the unsteady electroosmotic flow of linear Jeffreys fluids in microbial pipes. The Jeffreys fluid model for studying the rheology of fluids with ciliary motion was investigated by Ishtiaq et al. [53]. Ambreen et al. [54] focused on the heat transfer properties in oblique stagnation flow of Jeffreys fluids deformed by linear stretchable surfaces.

Finally, to the best of the authors’ knowledge, relatively minor EKEC efficiency research has been carried out on viscoelastic fluids, particularly on nanofluids with viscoelastic behavior. Accordingly, this article investigates the influences of periodic pressure gradients and magnetic fields with three distinct waveforms (cosine, triangle, square) on the flow potential, axial velocity, and EKEC efficiency of viscoelastic nanofluid. This is an oscillating problem, and oscillating pressure gradients can generate periodic pressures, which additionally generate periodic streaming potentials, resulting in time-evolving EKEC efficiency. By employing the Green function method, Gao et al. [55] studied the simple Maxwell model. In this paper, instead of using convective derivatives like the Oldroyd-B model, we investigated the Jeffreys model using time derivatives as a simpler linear model. The Jeffreys fluid model is a straightforward viscoelastic fluid model, and it is of great significance to study the viscoelasticity of polymer fluids. We consider non-steady state and related dimensionless parameters and visualize the impact of each dimensionless parameter on the fluid flow behavior within the microtubes. Especially, the analytic solution for axial velocity field is computed obtaining the Green’s function method. It is hoped these findings are able to offer relevant suggestions to the studies in microfluidics and provide some reference points for future research.

## 2. Materials and Methods 

The radius of a circular microtube is *R*_0_ and the length is *L* (Assuming *R*_0_ ≪ *L*). The two ends of the microtube are linked by two reservoirs, which are assumed to be infinitely vast in relation to the size of the microtube. A cylindrical coordinate system with axis *X* is established at the center of the microtube. There is a pressure difference Δ*p* = *p*_1_ − *p*_2_ between inlet and outlet. Viscoelastic nanofluids with Jeffreys fluid constitutive relation produce a uniform horizontal flow in the condition of the combined effect of evenly distributed magnetic field *B*_0_ in the *r* direction and a pressure gradient in the *X* direction flowing through a long circular microtube is considered for investigation. Fluids are considered unstable and unidirectional, which is considered as a symmetric (*z*:*z*) electrolyte, with *z* representing the valence of the electrolyte. At the same time, this is because of the chemical interaction between the electrolyte solution and the microtube wall forming an EDL at the solid–liquid interface. The excess free ions in the EDL migrate with the fluid, inducing a streaming current. Such currents can be applied to external electronic loads to achieve energy conversion and utilization. Furthermore, the movement of the fluid again causes the enhancement of the downstream potential of the microtube, forming a streaming potential opposite to the flow direction.

### 2.1. Analytical Solutions of the Velocity Field

As shown in Figure 1, consider viscoelastic nanofluids with Jeffreys properties subjected to axial pressure and lateral magnetic fluid forces in circular microtubules. The one-dimensional Cauchy momentum equation is illustrated as follows:(1)$$\rho_{eff}\frac{\partial U}{\partial T}=\frac{1}{R}\frac{\partial }{\partial R}\left(R\tau_{zR}\right)+\rho_{e}E_{s}-\frac{dP}{dx}-\sigma_{eff}B_{0}^{2}U,$$
where *T* is the time, *U* is the axial velocity, *τ_zR_* is the shear stress, *−dP*/*dx* = *P*_0_
*F*(*ΩT*) is the axial pressure gradient and *P*_0_ is the amplitude, *E_s_* is the streaming potential, *E_s_* = *E*_0_ *F*(*ΩT*), and *E*_0_ is the amplitude. Due to the elastic effect of the fluid, the periodic pressure gradient causes a periodic electric field; this behavior is called resonance [56]. *B*_0_ is the magnetic field intensity, *ρ_eff_*, *σ_eff_* is the effective density and effective electrical conductivity of the nanofluid, respectively; these are provided by the following equation:(2)$$\rho_{eff}=\varphi \rho_{s}+\left(1-\varphi \right)\rho_{f},$$
(3)$$\sigma_{eff}=\sigma_{f}\left(1+\frac{3\left(\sigma_{s}/\sigma_{f}-1\right)\varphi }{\left(\sigma_{s}/\sigma_{f}+2\right)-\left(\sigma_{s}/\sigma_{f}-1\right)\varphi }\right),$$
where *φ* is the volume fraction of the nanoparticles, *ρ_f_* is the density of fluid, *ρ_s_* is the density of a solid, *σ_s_* is electrical conductivity of nanoparticles, and *σ_f_* is electrical conductivity of base fluids.

Equation (1) is subject to the no-slip and symmetry boundary conditions
(4)$${U|}_{R=R_{0}}=0,{U|}_{T=0}=0,{U_{R}|}_{R=0}=0,U_{T}|{}_{T=0}=0.$$

The constitutive equation for the Jeffreys fluid model [50,57] is given by
(5)$$\left(1+\lambda_{1}\frac{\partial }{\partial T}\right)\tau_{zR}=\eta_{0}\left(\frac{\partial U}{\partial R}+\lambda_{2}\frac{\partial^{2}U}{\partial T\partial R}\right)$$
in Equation (5), *λ*_1_ and *λ*_2_ are the relaxation and retardation times, respectively; *η*_0_ = *µ_f_*/(1 − *φ*)^2.5^ *i*s the zero shear viscosity of the fluid; and *µ_f_* is the viscosity of the base fluid.

In order to solve the velocity field, we should obtain the EDL charge density firstly. The electric potential and local volume net charge density of EDL in quasi-steady state environment are described by the following Poisson–Boltzmann (PB) equations:(6)$$\frac{1}{R}\frac{\partial }{\partial R}\left(R\frac{d\psi }{dR}\right)=-\frac{\rho_{e}\left(R\right)}{\varepsilon },$$
(7)$$\rho_{e}\left(R\right)=ze\left(n_{+}-n_{-}\right),$$
where *ε* is the dielectric constant of the electrolyte solution, *e* is the elementary charge, *z* is the valence of ions, and *n*_+_ and *n*_−_ are the number densities of the electrolyte cations and anions, respectively, which is given by the Boltzmann distribution:(8)$$n_{\pm }=n_{0}\exp\left(\mp \frac{ez\psi \left(R\right)}{k_{B}T_{av}}\right)$$where *n*_0_ is the ion density of the bulk liquid, *T_av_* is the absolute temperature, and *k_B_* is the Boltzmann constant. In the meantime, we can use the Debye–Hückel approximation assuming that the potential is sufficiently small, i.e., sin*h*(*ψ*) *≈ ψ*, so we can obtain
(9)$$\frac{1}{R}\frac{\partial }{\partial R}\left(R\frac{d\psi }{dR}\right)=\kappa^{2}\psi ,\;\kappa ={\left(\frac{2n_{0}z^{2}e^{2}}{\varepsilon k_{B}T_{av}}\right)}^{\frac{1}{2}}.$$

The boundary conditions of the electrical potential are
(10)$$\psi {\left(R\right)|}_{R=R_{0}}=\psi_{0}\;,\;{\frac{\partial \psi \left(R\right)}{\partial R}|}_{R=0}=0,$$
where *ψ*_0_ is the wall zeta potential, *κ* is the Debye–Hückel parameter, and 1/*κ* denotes the thickness of the EDL. Solving Equations (9) and (10), the net charge density distribution can be expressed
(11)$$\rho_{e}\left(R\right)=-\varepsilon \kappa^{2}\psi_{0}\frac{I_{0}\left(kR\right)}{I_{0}\left({kR}_{0}\right)}.$$

*I*_0_ is the modified Bessel function of the first kind of order zero.

Introducing the following dimensionless group,
(12)$$r=\frac{R}{R_{0}},\;u=\frac{U}{u_{e}},\;u_{e}=-\frac{\varepsilon \varphi_{0}E_{x}}{\mu_{f}},\;K=\kappa R_{0},\;\gamma =\frac{\sigma_{f}}{\sigma_{eff}},\;t=\frac{\mu_{f}}{\rho_{f}R_{0}{}^{2}},\;\bar{E_{S}}=\frac{E_{S}}{E_{x}},\omega =\frac{\rho_{f}R_{0}{}^{2}}{\mu_{f}}\Omega ,\;Ha=B_{0}\sqrt{\frac{\sigma_{f}}{\mu_{f}}},\;\Psi =\frac{ez\psi }{k_{B}T_{av}},\;{\tilde{\tau }}_{zr}=\frac{\tau_{zR}R_{0}}{u_{e}\eta_{0}},De=\lambda_{1}\frac{\mu_{f}}{\rho_{f}R_{0}{}^{2}},{De}^{*}=\lambda_{2}\frac{\mu_{f}}{\rho_{f}R_{0}{}^{2}},$$
where *u_e_* is the Helmholtz–Smoluchowski electroosmotic velocity, *E_x_* is the characteristic electric field, and *K* is called electrokinetic width of the EDL denoting the ratio of radius *R*_0_ of microchannel to Debye length 1/*κ*. *ω* is the dimensionless frequency; Ha is the Hartmann number, which gives an estimate of the magnetic forces compared to the viscous forces. *De* and *De*^*^ are the Deborah numbers, which set the interplay between the characteristic relaxation time and retardation time of the fluid and viscous diffusion time of the flow.

After inserting dimensionless parameters (12) into Equations (1) and (5), the dimensionless momentum equation and constitutive equation have the following form:(13)$$\frac{\partial u}{\partial t}=-\frac{\rho_{f}\eta_{0}}{\rho_{eff}\mu_{f}}\frac{1}{r}\frac{\partial }{\partial r}\left(r{\tilde{\tau }}_{zr}\right)+\frac{\rho_{f}K^{2}{\overset{-}{E}}_{s}}{\rho_{eff}\mu_{f}}\frac{I_{0}\left(kr\right)}{I_{0}\left(k\right)}+\frac{\rho_{f}R_{0}{}^{2}P_{0}}{\rho_{eff}\mu_{f}u_{e}}F\left(\omega t\right)-\frac{\rho_{f}}{\rho_{eff}\gamma }{Ha}^{2}u$$
(14)$$\left(1+De\frac{\partial }{\partial t}\right){\tilde{\tau }}_{zr}=-\left(1+{De}^{*}\frac{\partial U}{\partial tT}\right)\frac{\partial u}{\partial r}$$

Replacing Equations (14) to (13) and introducing the simple form, we obtain that
(15)$$\begin{array}{l} \alpha \frac{\partial u}{\partial t}+De\frac{\partial^{2}u}{\partial t^{2}}-\beta_{1}\left(1+{De}^{*}\frac{\partial }{\partial t}\right)\left[\frac{1}{r}\frac{\partial }{\partial r}\left(r\frac{\partial u}{\partial r}\right)\right]-\beta_{2}{Ha}^{2}u\\ =\underset{Q\left(r,t\right)}{\underset{︸}{\beta K^{2}{\overset{-}{E}}_{s}\frac{I_{0}\left(kr\right)}{I_{0}\left(k\right)}+\beta_{3}De\frac{\partial F\left(\omega t\right)}{\partial t}\frac{I_{0}\left(kr\right)}{I_{0}\left(k\right)}+\beta_{4}F\left(\omega t\right)+\beta_{5}De\frac{\partial F\left(\omega t\right)}{\partial t}}} \end{array}$$
where $\beta =\frac{\rho_{f}}{\rho_{eff}}$, $\alpha_{1}=\frac{\eta_{0}}{\mu }$, $\beta_{1}=\alpha_{1}\cdot \beta$, $\beta_{2}=-\beta \cdot \frac{1}{\gamma }$, $u_{p}=\frac{R_{0}{}^{2}P_{0}}{\mu_{f}}$, $u_{r}=\frac{u_{p}}{u_{e}}$,$\beta_{3}=\beta E_{x}K^{2}$, $\beta_{4}=\beta \cdot u_{r}$ and $\alpha =1+\beta_{2}{DeHa}^{2}$, the initial conditions and relevant boundary conditions are as follows:(16)$${u|}_{t=0}=0,\;{u_{t}|}_{t=0}=0.{u|}_{r=1}=0,\;{u_{r}|}_{r=0}=0.$$

### 2.2. Resolution Procedure

Then, the velocity-analytic solution of Equation (15) is obtained using the Green’s function method, which can be expressed as follows:(17)$$\alpha \frac{\partial g}{\partial t}+De\frac{\partial^{2}g}{\partial t^{2}}-\beta_{1}\left(1+{De}^{*}\frac{\partial U}{\partial tT}\right)\left[\frac{1}{r}\frac{\partial }{\partial r}\left(r\frac{\partial u}{\partial r}\right)\right]-\beta_{2}{Ha}^{2}g=\frac{\delta \left(r-l\right)\delta \left(t-\tau \right)}{2\pi r},\;0<r,l<1,\;t,\tau >0.$$

The initial conditions and boundary conditions are as follows:(18)$$g\left(1,t|l,\tau \right)=0,g\left(r,0|l,\tau \right)=0,g_{r}\left(0,t|l,\tau \right)=0,g_{t}\left(r,0|l,\tau \right)=0,$$
where Δ(*x*) is the Dirac delta function. 

Then, we change *r*, *t* to *l*, *τ* in Equations (15) and (17):(19)$$\alpha \frac{\partial u}{\partial \tau }+De\frac{\partial^{2}u}{\partial \tau^{2}}-\beta_{1}\left(1+{De}^{*}\frac{\partial }{\partial \tau }\right)\left[\frac{1}{l}\frac{\partial }{\partial l}\left(l\frac{\partial u}{\partial r}\right)\right]-\beta_{2}{Ha}^{2}u=Q\left(l,\tau \right),$$
(20)$$\alpha \frac{\partial g}{\partial \tau }+De\frac{\partial^{2}g}{\partial \tau^{2}}-\beta_{1}\left(1+{De}^{*}\frac{\partial }{\partial \tau }\right)\left[\frac{1}{l}\frac{\partial }{\partial l}\left(l\frac{\partial u}{\partial l}\right)\right]-\beta_{2}{Ha}^{2}u=\frac{\delta \left(r-l\right)\delta \left(t-\tau \right)}{2\pi l}$$

Equation (19) is multiplied by *g*, and Equation (20) is multiplied by *u*. Finally, the two equations are subtracted and integrated. Using the following simplified formula, we can obtain the dimensionless velocity:(21)$$u\left(r,t\right)=\int_{0}^{t}\int_{0}^{1}Q\left(l,\tau \right)g\left(r,t;l;\tau \right)dld\tau .$$

Then, let us solve for *g*, and we first apply Laplace derivative theorem to Equation (17) to obtain:(22)$$\left(\alpha s+{Des}^{2}-\beta_{2}{Ha}^{2}\right)\bar{g}-\left(\beta_{1}+s\beta_{1}{De}^{*}\right)\left(\frac{\partial^{2}\bar{g}}{\partial r^{2}}+\frac{\partial \bar{g}}{\partial r}\right)=\frac{\delta \left(r-l\right)}{2\pi r}e^{-s\tau }.$$

Then, the Hankel transformation of Equation (22) is as follows:(23)$$\left(\beta_{1}k_{i}^{2}+s\beta_{1}{De}^{*}k_{i}^{2}+\alpha s+{Des}^{2}-\beta_{2}{Ha}^{2}\right)\tilde{\bar{g}}=\frac{J_{0}\left({lk}_{i}\right)}{2\pi }e^{-s\tau },$$
namely the equation of
(24)$$\tilde{\bar{g}}=\frac{J_{0}\left({lk}_{i}\right)}{2\pi }\frac{1}{De}D_{2}^{-\frac{1}{2}}\frac{e^{-s\tau }D_{2}^{\frac{1}{2}}}{{\left[s-\left(-D_{1}\right)\right]}^{2}-D_{2}},$$
where $D_{1}=\frac{\alpha +\beta_{1}De^{*}k_{i}^{2}}{2De}$, $D_{2}=\frac{\beta_{1}k_{i}^{2}-\beta_{6}}{De}-D_{1}^{2}$.

The inverse Laplace transform of Equation (24) can be written as
(25)$$\tilde{g}\left(k_{i},t;l,\tau \right)=\frac{J_{0}\left({\mathrm{lk}}_{i}\right)}{2\pi }\frac{1}{De}D_{2}^{-\frac{1}{2}}H\left(t-\tau \right)e^{-D_{1}\left(t-\tau \right)}\sin\left[D_{2}^{\frac{1}{2}}\left(t-\tau \right)\right].$$

Applying the inverse Hankel transform to Equation (25), we obtain
(26)$$g\left(r,t;l,\tau \right)=\frac{1}{\pi De}\sum_{i=1}^{\infty }J_{0}\left({lk}_{i}\right)D_{2}^{-\frac{1}{2}}H\left(t-\tau \right)e^{-D_{1}\left(t-\tau \right)}\sin\left[D_{2}^{\frac{1}{2}}\left(t-\tau \right)\right]\frac{J_{0}\left({rk}_{i}^{2}\right)}{J_{1}^{2}\left(k_{i}\right)}$$

We get *u* by substituting Equation (26) into Equation (21). Then, we introduce three different periodic functions. 

If the cosine wave is a periodic function,
(27)F(ωt)=cos(ωt).

The velocity profile will be gained by Equation (21) as
(28)$$u\left(r,t\right)=\frac{1}{\pi {De}_{}}\sum_{i=1}^{\infty }\left\{D_{2}^{-\frac{1}{2}}\frac{J_{0}\left({rk}_{i}^{2}\right)}{J_{1}^{2}\left(k_{i}\right)}\left[L_{1}\times L_{7}-L_{2}\times L_{8}\right]\right\}.$$

The corresponding exact results for *L*_1_, *L*_2_, *L_7_* and *L_8_* are also listed in Appendix A.

If the triangular waveform is a periodic function,
(29)$$F\left(\omega t\right)=\frac{8}{\pi^{2}}\sum_{m=1}^{\infty }\frac{\sin\left(\frac{m\pi }{2}\right)}{m^{2}}\sin\left(m\omega t\right),$$

The dimensionless axial velocity will be obtained from Equation (21) as
(30)$$u\left(r,t\right)=\frac{8}{\pi^{3}De}\sum_{i=1}^{\infty }\left\{D_{2}^{-\frac{1}{2}}\frac{J_{0}\left({rk}_{i}^{2}\right)}{J_{1}^{2}\left(k_{i}\right)}\sum_{m=1}^{\infty }\left(\frac{\sin\left(\frac{m\pi }{2}\right)}{m^{2}}\left(L_{1}\times L_{13}\right)+\frac{\sin\left(\frac{m\pi }{2}\right)}{m}\left(L_{2}\times L_{14}\right)\right)\right\}.$$

In addition, the expressions for *L*_13_ and *L*_1*4*_ are also listed in Appendix A.

If the square waveform is a periodic function,
(31)$$F\left(\omega t\right)=\frac{2}{\pi }\sum_{m=1}^{\infty }\frac{1-\cos\left(m\pi \right)}{m}\sin\left(m\omega t\right).$$

The dimensionless axial velocity will be obtained from Equation (21) as follows:(32)$$u\left(r,t\right)=\frac{2}{\pi De}\sum_{i=1}^{\infty }\left\{D_{2}^{-\frac{1}{2}}\frac{J_{0}\left({rk}_{i}^{2}\right)}{J_{1}^{2}\left(k_{i}\right)}\sum_{m=1}^{\infty }\left(\frac{1-\cos\left(m\pi \right)}{m}\left(L_{1}\times L_{13}\right)+\left(1-cos\left(m\pi \right)\right)\left(L_{2}\times L_{14}\right)\right)\right\}.$$

### 2.3. Analytical Solution of the Streaming Potential

In order to maintain the electrical neutrality of the fluid system, the conduction current and the streaming current are kept in balance and are in a steady state: (33)$$I=2\pi ez\int_{0}^{ℜ}\left(n_{+}u_{+}-n_{-}u_{-}\right)rdr=I_{s}+I_{c}=0,$$
where *I_c_* is the conduction current, *I_s_* is the streaming current, and the combination of viscoelastic fluid electromigrative velocity and advection velocity are *u_±_* is as follows:(34)$$u_{\pm }=U\pm \frac{{ezE}_{s}}{f},$$
where *f* is the ionic friction coefficient. We can substitute Equations (8) and (34) into Equation (33); then, the form is as follows:(35)$$\int_{0}^{1}ru\frac{I_{0}\left(Kr\right)}{I_{0}\left(K\right)}dr=\frac{{ez\;E\;¯}_{s}E_{x}}{2u_{e}\psi_{0}f}.$$

Substitute Equation (28) into Equation (35), and the dimensionless streaming potential of the cosine waveform is as follows:(36)$${\overset{-}{E}}_{s}=\frac{\frac{1}{\pi De}\sum_{i=1}^{\infty }\left\{D_{2}^{-\frac{1}{2}}\int_{0}^{1}r\frac{J_{0}\left({rk}_{i}^{2}\right)}{J_{1}^{2}\left(k_{i}\right)}\frac{I_{0}\left(Kl\right)}{I_{0}\left(K\right)}dr\left[L_{1}\times L_{7}-L_{2}\times L_{8}\right]\right\}}{A},$$
where $A=\frac{ezE_{x}}{2u_{e}\psi_{0}f}$.

Substitute Equation (30) into Equation (35), and the dimensionless streaming potential of triangular waveform is as follows:(37)$${\overset{-}{E}}_{s}=\frac{\frac{8}{\pi^{3}De}\sum_{i=1}^{\infty }\left\{D_{2}^{-\frac{1}{2}}\int_{0}^{1}r\frac{J_{0}\left({rk}_{i}^{2}\right)}{J_{1}^{2}\left(k_{i}\right)}\frac{I_{0}\left(Kl\right)}{I_{0}\left(K\right)}dr\sum_{m=1}^{\infty }\left(\frac{\sin\left(\frac{m\pi }{2}\right)}{m^{2}}\left(L_{1}\times L_{13}\right)+\frac{\sin\left(\frac{m\pi }{2}\right)}{m}\left(L_{2}\times L_{14}\right)\right)\right\}}{A}.$$

Substitute Equation (32) into Equation (35), we shall finally get the dimensionless streaming potential of the square waveform as follows:(38)$${\overset{-}{E}}_{s}=\frac{\frac{2}{\pi De}\sum_{i=1}^{\infty }\left\{D_{2}^{-\frac{1}{2}}\int_{0}^{1}r\frac{J_{0}\left({rk}_{i}^{2}\right)}{J_{1}^{2}\left(k_{i}\right)}\frac{I_{0}\left(Kl\right)}{I_{0}\left(K\right)}dr\sum_{m=1}^{\infty }\left(\frac{1-\cos\left(m\pi \right)}{m}\left(L_{1}\times L_{13}\right)+\left(1-\cos\left(m\pi \right)\right)\left(L_{2}\times L_{14}\right)\right)\right\}}{A}.$$

### 2.4. Electric Energy Conversion Efficiency

During the generation of the streaming electric field (*E_s_*) and the streaming current (*I_s_*), the mechanical energy of the pressure-driven flow is transformed to the electric energy. The EKEC efficiency *ξ* meaning is as follows: (39)$$\xi =\left|\frac{P_{\mathrm{out}}}{P_{\mathrm{in}}}\right|,$$where *P_in_* and *P_out_* respectively represent the output and input power, which are defined as
(40)$$P_{\mathrm{out}}=\left|\left(\frac{I_{s}}{2}\right)\left(\frac{E_{s}}{2}\right)\right|,$$
(41)$$P_{\mathrm{in}}=\left|-\frac{dp}{dx}Q_{\mathrm{in}}\right|,$$
where *Q_in_* indicates the rate of flow for this pressure driven flow, and the expressions are
(42)$$Q_{\mathrm{in}}=2\pi \int_{0}^{1}-\frac{1}{4\mu_{f}}\frac{dP}{dx}\left(R_{0}{}^{2}-r^{2}\right)rdr,$$
(43)$$I_{s}=2\pi ez\int_{0}^{1}u\left(n_{+}-n_{-}\right)rdr.$$

As a result, the *ξ* derived from Equations (39)–(43) can be written as
(44)$$\xi =\frac{4e^{2}z^{2}{\bar{E}}_{s}^{2}E_{x}^{2}n_{0}\mu_{f}}{p_{0}^{2}F^{2}\left(\omega t\right)R_{0}{}^{2}f}.$$

Substituting Equation (27) into Equation (44), we obtain the EKEC efficiency of cosine waveform as
(45)$$\xi =\frac{4e^{2}z^{2}n_{0}{\bar{E}}_{s}^{2}E_{x}^{2}\mu_{f}}{p_{0}^{2}\cos^{2}\left(\omega t\right){ℜ}^{2}f}.$$

Substituting Equation (29) into Equation (44), we obtain the EKEC efficiency of triangular waveform as
(46)$$\xi =\frac{\pi^{4}e^{2}z^{2}n_{0}{\bar{E}}_{s}^{2}E_{x}^{2}\mu_{f}}{{16p}_{0}^{2}{\left(\frac{8}{\pi^{2}}\sum_{m=1}^{\infty }\frac{\sin\left(\frac{m\pi }{2}\right)}{m^{2}}\sin\left(m\omega t\right)\right)}^{2}{ℜ}^{2}f}.$$

Substituting Equation (31) into Equation (44), we obtain the EKEC efficiency of square waveform as
(47)$$\xi =\frac{\pi^{2}e^{2}z^{2}n_{0}{\bar{E}}_{s}^{2}E_{x}^{2}\mu_{f}}{p_{0}^{2}{\left(\frac{2}{\pi }\sum_{m=1}^{\infty }\frac{1-\cos\left(m\pi \right)}{m}\cos\left(m\omega t\right)\right)}^{2}{ℜ}^{2}f}.$$

## 3. Results and Discussion

In the above discussion, we derived analytical solutions for velocity, streaming potential, and EKEC efficiency of viscoelastic nanofluids with Jeffreys constitutive relation on the condition of the combined action of a transversal magnetic field and axial pressure gradient in a microtube. The analytical solutions depend largely on many of the dimensionless parameters defined above, such as Deborah numbers *De* and *De*^*^, dimensionless frequency *ω*, Hartmann number *Ha*, volume fraction of nanoparticles *φ*, and electro-kinetic width *K*. We first need to give the practical reference range of these dimensionless parameters in practical problems according to the following related physical variable quantities: *R*_0_ = 100 µm, *ε* = 7 × 10^−10^ *C*^2^·N^−1^·m^−2^, *e* = 1.6 × 10^−19^ *C*, *n*_0_ = 10^−5^ m^−3^, *k_B_* = 1.381 × 10^−23^ J·K^−1^,*T_av_* = 298 K, *ψ*_0_ = −0.025 V, *z* = 1, *μ_f_* = 8.91 × 10^−3^ kg·m^−1^·s^−1^, *ρ_s_* = 3600 kg·m^−3^, *ρ_f_* = 997.1 kg·m^−3^, *σ_s_* = 10^−12^ S·m^−1^, *f* = 10^−12^ N·s·m^−1^, *σ_f_* = 0.05 S·m^−1^. The relaxation time *λ*_1_ ranges from 10^−4^ s to 10^−2^ s, and the retardation time *λ*_2_ is generally smaller than the relaxation time. On the basis of *De* = $\frac{\mu_{f}}{\rho_{f}R_{0}^{2}}$, the range of *De* is 0 to 10, and the range of *De*^*^ is 0 to 10. The range of Ha is set from 0 to 3, and the corresponding range of the applied magnetic field *B*_0_ is 40 *mT* to 0.44 *T*; according to *Ha* = *B*_0_$\sqrt{\frac{\sigma_{f}}{\mu_{f}}}$, the range of *ω* varies from 1 to 200, and the range of *φ* changes from 0.00 to 0.05 and the range of *K* from 1 to 10 owing to *K* = *κR*_0_; these conform to accept the value of physical research.

In this paper, the streaming potential and EKEC efficiency under the combined action of the axial pressure gradient and transverse magnetic field are investigated in the Jeffreys fluid model. It is worth noting that Gao et al. [55] performed a similar analysis to the current work, except that they assumed a Maxwell fluid. Where λ_1_ ≠ 0, λ_2_ ≠ 0, if we set λ_1_ = 0, the Jeffreys model in this paper becomes the Maxwell model. While keeping other parameters consistent, this paper compares the results of the existing study with that of Gao et al. [55] in Figure 2. Figure 2a is the comparison of the streaming potential under a cosine wave, and Figure 2b is the comparison of EKEC efficiency under the action of a square wave. It can be seen from Figure 2 that the streaming potential and the EKEC efficiency are basically consistent with the research results of Gao et al. [55].

Figure 3 shows the distribution of dimensionless axial velocity in the case of the cosine waveform for the various values of *De*^*^ when other parameters are fixed. The values of *De*^*^ in Figure 3 are set to 0.5, 2.0, 3.0, and 5.5, respectively. *De** is defined as the rate between the characteristic retardation time of the fluid and viscous diffusion time of the flow. It is interesting to note that the absolute magnitude of the dimensionless velocity declines as the value of the *De*^*^ increases, which is the opposite of the impacts of *De* on the velocity field. The increase in the value of *De*^*^ indicates that the retardation time becomes larger, and the speed reduces with the increase of the *De*^*^ because the retardation time tends to inhibit the motion of the fluid. In the same way, the dimensionless velocity changes relatively gently inside the microtube, while the velocity fluctuates with time near the microchannel wall and Changes more sharply than in the middle of the microtube.

Figure 4 shows the variety of dimensionless velocity with dimensionless time under various values of the dimensionless frequencies of the triangular waveform. The values of dimensionless frequencies are 30, 60, 90, and 120, respectively. When the pressure gradient is a superimposed waveform, it can be seen from Figure 4 that the overall decrease of the dimensionless velocity is due to the increase of the dimensionless frequency. Results show that a larger dimensionless frequency is due to a lesser time period; meanwhile, the velocity spread time becomes shorter and the oscillation becomes quicker, the fluid movement cannot progressive sufficiently, and the fluid momentum does not have enough time to diffuse into the volume flow. As a result, the overall dimensionless velocity is decreasing. For a fixed *ω*, it can be seen that the dimensionless velocity on the surface of the microtubule wall is higher than that inside of the microtubule. The dissipation time scale is much larger than the vibration period, and the fluid velocity only changes near the two walls, so the fluid motion does not have enough time to diffuse to the plane in the middle of the microtube. It is also worth mentioning that the dimensionless axial velocity exhibits a simple harmonic motion with time.

In Figure 5, we portray the dimensionless velocity distribution for the various *De*^*^ in the condition of a square waveform in the microtube. The values of *De*^*^ are set to 0.5, 1.0, 3.0, and 5.5, respectively. As expected, it is observed from Figure 5 that the dimensionless velocity reduces with the rise in *De*^*^, and the wall velocity is bigger than that in the central area from a single image, which is similar to the result displayed in Figure 3. Because a longer retardation time induces a greater flow resistance, the amplitude of the flow velocity decreases, and then the velocity will decrease. Velocity gradients occur only near the wall, momentum diffuses only in the boundary layer and away from the wall, and the flow is less affected by oscillatory effects. Different from Figure 3, the periodic variation of velocity with time approximates harmonics due to the pressure gradient of the superimposed waveform.

Figure 6 illustrates the effects of the dimensionless streaming potential of the cosine waveform with the dimensionless time of several dimensionless parameters (*De*, *De*^*^, *K*, *ω*). From Figure 6a, the dimensionless streaming potential as a function of dimensionless time at different values of *De* in the microtubule can be seen. The values of *De* are set to 1.0, 2.5, 4.5, and 6.0, respectively. It can be interestingly observed from Figure 6a that, with the further increase of *De*, the absolute values of dimensionless streaming potential start to gain; for a fixed *De*, dimensionless streaming potential fluctuates with time. The increase of *De* signifies that the relaxation time increases, which means that the fluid has a large elastic action leading to huge velocity amplitude, and then leads to an increase in the absolute value of dimensionless streaming potential. From a physical point of view, this means that the diffusion time is much larger than the characteristic time of the oscillating electric field.

Figure 6b portrays an effect of the variation of dimensionless streaming potential of dimensionless time for several values of *De*^*^ (*De*^*^ = 0.5, 1.0, 3.0, and 5.5, respectively) in cosine waveform. In complete contrast to Figure 6a, the absolute value of the dimensionless streaming potential decreases due to the increase of value of *De*^*^, and the same thing is that it fluctuates up and down periodically over time. A cosine wave is the simplest periodic pressure gradient that drives the periodic motion of a fluid. However, due to the inhibitory effect of the retardation time, the velocity reduces with the increase of the retardation time, and the flow becomes slower and slower, which causes reduction in the absolute value of the streaming potential.

Figure 6c portrays the variation of dimensionless streaming potential of dimensionless time for different dimensionless electrokinetic widths (*K* = 2, 3, 4, and 5, respectively) under the cosine waveform. Dimensionless K represents the ratio of the radius of the microtubule to the Debye length. From Figure 6c, it is clearly shown that the rise of dimensionless K leads to the decline of the absolute value of the dimensionless streaming potential, and for a single dimensionless *K*, the dimensionless streaming potential fluctuates with time, and the change tends to be steady with the increase of time. The lesser value of *K* results in the enormous EDL, signaling that the density of free charged ions in the microtube is larger, so the resistance to fluid motion is bigger, which results in the greater degree of fluid motion fluctuation and incomplete development of fluid motion.

Figure 6d depicts the dimensionless streaming potential distribution on the dimensionless time for different dimensionless frequency values under the cosine wave. The values of *ω* are 30, 60, 90, and 120, respectively. With the increase of the value of dimensionless *ω*, the oscillation period of the dimensionless streaming potential results in a reduction, and the amplitude turns to smaller. It can be explained that the larger the dimensionless frequency, the much larger the diffusion time period than the induced electric field period, so the faster the fluid oscillates and the dimensionless streaming potential decreases. Under the condition of cosine waveform, it can also be noted that the streaming potential of the fluid has a buffering stage at the original moment, and ultimately the variation of the dimensionless streaming potential with growing time is curved and stable.

Figure 7a exhibits the effects of the dimensionless streaming potential of triangular waveform with respect to the Deborah number *De* with some values of dimensionless time. The values of *De* are set to 1.0, 2.5, 4.5, and 6.0, respectively. As seen in Figure 7a, the dimensionless streaming potential increases with the increase in the *De* number. As a result, the larger *De* is due to the larger relaxation time, and the elastic action of the fluid plays a dominant role, ultimately leading to the increase of the dimensionless streaming potential. In the condition of the triangular wave, the buffering phase of the fluid flow potential is comparatively strong at the original moment and then with the increase of time, the streaming potential as a whole will be stable and comparable to simple harmonic changes.

Figure 7b illustrates the dimensionless streaming potential at several values of the Deborah number *De*^*^ in the case of triangular wave. The values of *De*^*^ are set to 0.5, 1.0, 3.0, and 5.5, respectively. It is evident that the increase in the *De*^*^ numbers leads to the overall decrease in the streaming potential; this is in contrast to Figure 7a. It can be explained that a longer retardation time indicates that the fluid is more viscous, which leads to greater flow resistance that obstructs the motion of fluid. The same as Figure 7a is that of the increase of time; the trend of the dimensionless streaming potential will modify regularly like simple harmonics. In summary, it can be concluded that the increase of *De* promotes the streaming potential, while the increase of *De*^*^ inhibits the streaming potential.

Figure 7c depicts the effect of dimensionless streaming potential for several dimensionless electro-kinetic widths with dimensionless time in triangular waveform with other parameters unchanged. The dimensionless *K* values are set to be 2, 3, 4, and 5, respectively. We can clearly see that the change of dimensionless streaming potential decreases and becomes more moderate as the dimensionless electro-kinetic widths expand. The larger the dimensionless electro-kinetic width *K* indicates that the EDL thickness is narrower. It demonstrates that the smaller the density of free charged ions in the microchannel, the smoother the fluctuation of the fluid movement and the smaller the streaming potential.

Figure 7d exhibits variation of the dimensionless streaming potential with dimensionless time of the dimensionless frequency under triangular wave. The values of *ω* are set as 30, 60, 90, and 120, respectively. We can clearly find that the streaming potential decreases with the increase in the dimensionless frequency. This is because the greater dimensionless frequency makes the fluid motion oscillation period smaller and the fluid motion is not adequately developed, making the fluid motion hindered and hence the streaming potential reduced. In addition, we can see from Figure 7d that, in the condition of triangular waveform periodic pressure gradient, the dimensionless streaming potential changes in a harmonic shape with the increase of time.

The EKEC efficiency variation for several Deborah number *De* with time under the triangular waveform in a microtube is portrayed in Figure 8a. De is set to set to 1.0, 2.5, 4.5, and 6.0, respectively. It can be seen from Figure 8a that the Deborah number *De* is straightforward for inducing the huge EKEC efficiency. Since the elasticity of the fluid becomes meaningful with the increase of the relaxation time, the EKEC efficiency amplitude likewise increases with the increase of the relaxation time. A longer relaxation time means a larger elastic effect and a smaller recovery capacity, resulting in larger velocity amplitude, which in turn leads to an increase in the EKEC efficiency. The elasticity of viscoelastic fluids has a great influence on electro-kinetic energy conversion.

The variation of EKEC efficiency with time for various Deborah number *De*^*^ (There, *De*^*^ = 0.5, 1.0, 3.0, and 5.5, respectively) in the triangular wave is demonstrated in Figure 8b. It can be noted that the EKEC efficiency declines with the increase of Deborah number *De*^*^. The proposed reason for that *De*^*^ is a major parameter reflecting the viscosity of the fluid. Viscoelastic fluid reflects a huge viscous influence when *De*^*^ is improved, which means that the fluid has a large viscous effect leading to a small velocity amplitude. With the increase of *De*^*^, the retardation time is much greater than the excitation oscillation period, and the fluid will not have enough time to adapt to the change of the oscillating flow or the reduction of EKEC efficiency.

Figure 8c refers to the variation of the EKEC efficiency at various values of dimensionless electro-kinetic widths (*K* = 2, 3, 4, and 5, respectively) in the case of triangular waveform. From Figure 8c, we can clearly see that a larger electric width leads to a smaller EKEC efficiency. As expected, as the dimensionless *K* expands, this means that the width of the EDL is shrinking, which leads to a decrease in the number of EDL ions, thus dropping the efficiency of the EKEC. EDL width is bitty, the greater the density of free charged ions in the microchannel, the greater the resistance of fluid movement, leading to the fluid motion wave being more and more gradual. Consequently, the bigger *K* turns more smoothly to the fluctuation of EKEC efficiency.

Figure 8d portrays the variation of the EKEC efficiency with dimensionless time for different dimensionless frequency. It can be clearly seen that the magnitude of EKEC efficiency increases with the increase of dimensionless frequency. The results show that the EKEC efficiency peaks first when the dimensionless *ω* increases. The reason is that, at high frequencies, the oscillation period shortens as the dimensionless ω increases. Therefore, EKEC efficiency at high frequencies (*ω* = 120) will be maximized preferentially. On the other hand, when the dimensionless ω decreases, the oscillation period increases and the time reaching the peak becomes too long.

Figure 9a displays the change of the EKEC efficiency with dimensionless time for some Deborah number *De* (*De* = 1.0, *De* = 2.5, *De* = 4.5, and *De* = 6.0, respectively) in the condition of square waveform. It can be observed interestingly from Figure 9a that, with the further increase of Deborah number *De*, the EKEC efficiency starts to increase as a whole. Under the conditions of the pressure gradient and low frequency of the square wave waveform, there is a large initial phase, so the EKEC rose sharply and then diminishes at the initial moment, and then shows a slight fluctuation and an overall upward trend with time. When *De* progressively increases, the oscillation period is gently approached and smaller than the relaxation time, resulting in the shortening of the time required for the fluid to regulate oscillations, so EKEC efficiency increases.

Figure 9b presents the effect of EKEC efficiency about several Deborah numbers *De*^*^ for the square waveform with dimensionless time. *De*^*^ is set to 0.5, 1.0, 3.0, and 5.5, respectively. The results show that the EKEC efficiency decreases due to the increase of Deborah numbers *De*^*^; this result is in contrast to Figure 9a. However, the overall trend of EKEC efficiency is the same, since it has the conditions of square waveform and low frequency. Now the flow of the fluid is hardly influenced by the oscillation because the large viscous force of the fluid will induce a substantial flow resistance; therefore, the chemical energy generated in the EDL and the mechanical energy generated in the fluid cannot be entirely converted into electrical energy, resulting in a significant decrease in the efficiency of the EKEC. Improving the EKEC efficiency of the fluid can appropriately reduce the viscosity of the fluid.

Figure 9c depicts the influence of EKEC efficiency with time for the several values of Hartmann numbers (*Ha* = 0.5, *Ha* = 1.0, *Ha* = 2.0 and *Ha* = 3.0, respectively) under the square waveform. The dimensionless Hartmann number reflects the force of the magnetic field. From this figure, we can obviously find that the value of EKEC efficiency increases as the Hartmann numbers gradually increase in the condition of the superimposed square waveform, which means that, along with the increase of magnetic field force, the electric energy conversion in the fluid is excellent. In particular, when the dimensionless Ha value is 1, the EKEC efficiency shows an obvious fluctuation trend with time due to the periodic pressure.

We portray the variation of EKEC efficiency *ξ* with dimensionless time for different volume fractions of nanoparticles in the case of square waveform in Figure 9d. The *φ* is set to be 0.00, 0.02, 0.03, and 0.05, respectively. We can observe interestingly in Figure 9d that the range of the EKEC efficiency reduces as the value of the volume fraction of nanoparticles is enhanced. This is because the effective viscosity of the fluid can be improved with the increase of the volume fraction of nanoparticles. Nevertheless, an increase in the number of nanoparticles will undermine the velocity of the fluid, resulting in a reduction in streaming potential, which in turn leads to a decrease in EKEC efficiency. In addition, due to the periodic pressure gradient, the overall EKEC efficiency fluctuates with the passage of time.

Figure 10 depicts the EKEC efficiency of ordinary Newtonian nanofluid (*λ*_1_ = *λ*_2_ = 0), Jeffreys nanofluid (*λ*_1_ ≠ 0, *λ*_2_ ≠ 0), and Maxwell nanofluid (*λ*_1_ ≠ 0, *λ*_2_ = 0) in microtube as a function of dimensionless time. In Figure 10, EKEC efficiency changes with time and the peak value becomes larger. Moreover, the EKEC efficiency of viscoelastic fluid is evidently superior to that of ordinary Newtonian fluid, which is due to the shear-thinning effect (higher fluidity). Due to the obstruction of retardation time in Jeffreys fluid, the EKEC efficiency of Maxwell fluid is superior to that with Jeffreys fluid. It can be noted that the viscoelastic fluid can greatly improve the EKEC efficiency compared with the Newtonian fluid. In summary, it is valuable to study the EKEC efficiency of viscoelastic fluids in a micro-electro-mechanical system.

## 4. Conclusions

In this paper, the analytical solutions of axial velocity, streaming potential, and EKEC efficiency of viscoelastic fluids under three different periodic axial pressure gradients and transverse magnetic fields are derived in unsteady state. The effects of some dimensionless parameters (*De*, *De*^*^, *K*, *Ha*, *ω*, *φ*) on the axial velocity, flow potential, and EKEC efficiency are discussed and graphically shown. These results demonstrate that the velocity variation is snugly restricted to a narrow area against the EDL. *De* promotes the streaming potential and EKEC efficiency of a microtube, while *De*^*^ inhibits them. The increase of dimensionless *ω* will reduce the amplitude of fluid flow and shorten the oscillation period. Consequently, the EKEC efficiency and streaming potential are able to be enhanced by tuning the dimensionless parameters. The fluids flow trend of square wave and triangle wave is similar and different from that of the cosine wave. More interestingly, the EKEC efficiency of viscoelastic fluid is obviously superior to that of the Newtonian fluid, and the EKEC efficiency of the Maxwell fluid appears to be an advantage corresponding to the Jeffreys fluid.

## Figures and Tables

**Figure 1 nanomaterials-12-03355-f001:**
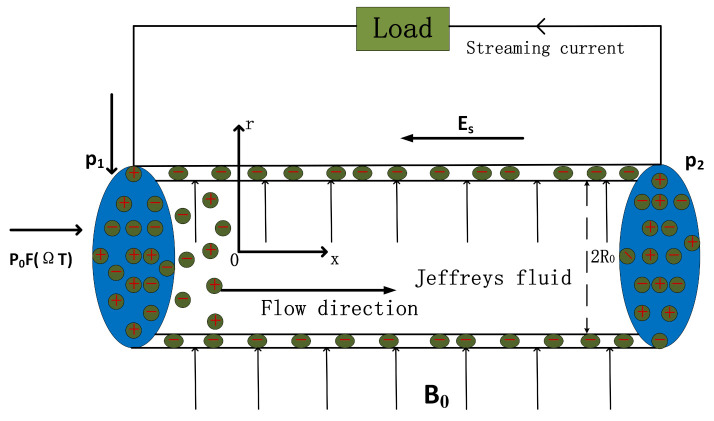
A sort of EKEC system.

**Figure 2 nanomaterials-12-03355-f002:**
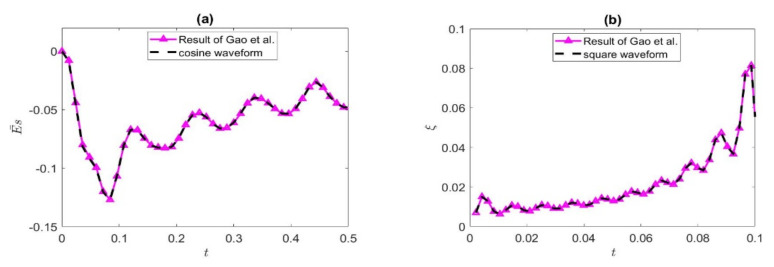
The present results are compared with the streaming potential (**a**) and EKEC efficiency (**b**) of Gao et al. [55] in (**a**) *De* = 1, *φ* = 2%, *ω* = 60, *Ha* = 2, *K* = 2; (**b**) *De* = 1, *φ* = 2%, *ω* = 2, *Ha* = 1, *K* = 1.5.

**Figure 3 nanomaterials-12-03355-f003:**
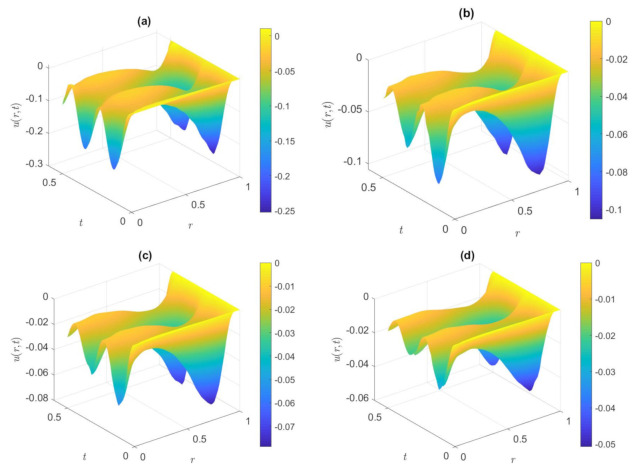
Dimensionless axial velocity variation of the cosine waveform at (**a**) *De*^*^ = 0.5, (**b**) *De*^*^ = 2.0, (**c**) *De*^*^ = 3.0, (**d**) *De*^*^ = 5.5. (*De* = 6, *φ* = 1%, *ω* = 30, *Ha* = 2, *K* = 6).

**Figure 4 nanomaterials-12-03355-f004:**
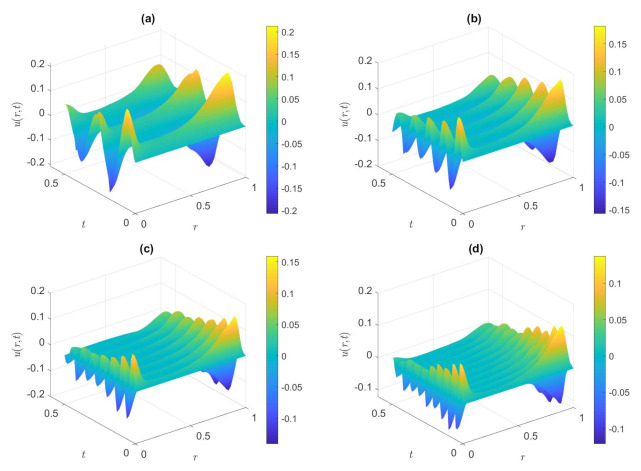
Dimensionless axial velocity variation of the triangular waveform in the microctube at (**a**) *ω* = 30, (**b**) *ω* = 60, (**c**) *ω* = 90, (**d**) *ω* = 120. (*De* = 1, *De*^*^ = 0.2π, *φ* = 2%, *Ha* = 2, *K* = 8).

**Figure 5 nanomaterials-12-03355-f005:**
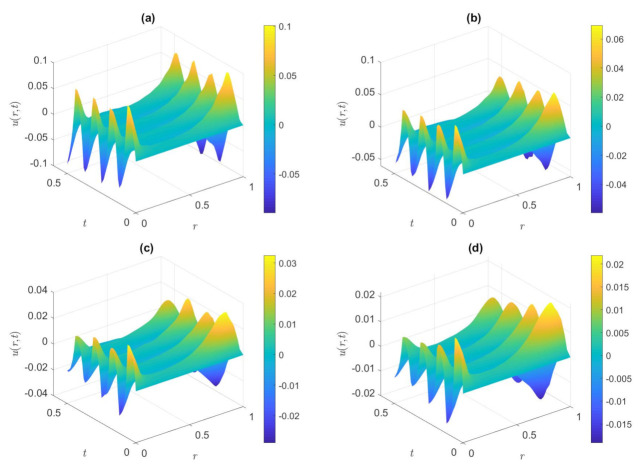
Dimensionless axial velocity distribution of the square waveform in the microtube at (**a**) *De*^*^ = 0.5, (**b**) *De*^*^ = 1.0, (**c**) *De*^*^ = 3.0, (**d**) *De*^*^ = 5.5. (*De* = 6, *φ* = 2%, *ω* = 50, *Ha* = 2, *K* = 8).

**Figure 6 nanomaterials-12-03355-f006:**
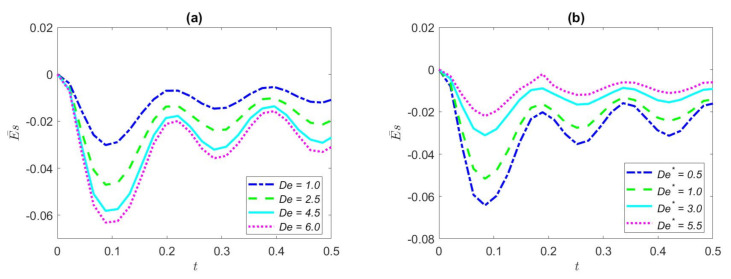
Effects of the dimensionless streaming potential of the cosine waveform for some values of *De*, *De**, *K*, *ω.* (**a**) *De** = 0.5, *φ* = 2%, *ω* = 35, *Ha* = 2, *K* = 3; (**b**) *De* = 5.6, *φ* = 2%, *ω* = 40, *Ha* = 2, *K* = 3; (**c**) *De* = 3, *De** = 0.5, *φ* = 2%, *ω* = 40, *Ha* = 2; (**d**) *De* = 1.5, *De** = 0.5, *φ* = 2%, *Ha* = 1, *K* = 3.

**Figure 7 nanomaterials-12-03355-f007:**
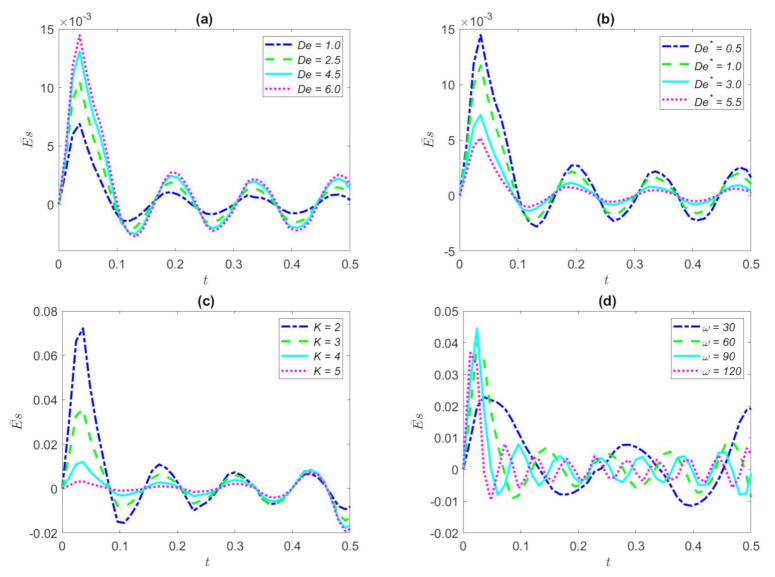
Effects of the dimensionless streaming potential of the triangular waveform for some values of *De*, *De*^*^, *K*, *ω.* (**a**) *De*^*^ = 0.5, *φ* = 2%, *ω* = 45, *Ha* = 2, *K* = 2; (**b**) *De* = 6, *φ* = 2%, *ω* = 45, *Ha* = 2, *K* = 2; (**c**) *De* = 1, *De*^*^ = 0.5, *φ* = 2%, *ω* = 50, *Ha* = 2; (**d**) *De* = 1, *De*^*^ = 0.5, *φ* = 2%, *Ha* = 2, *K* = 3.

**Figure 8 nanomaterials-12-03355-f008:**
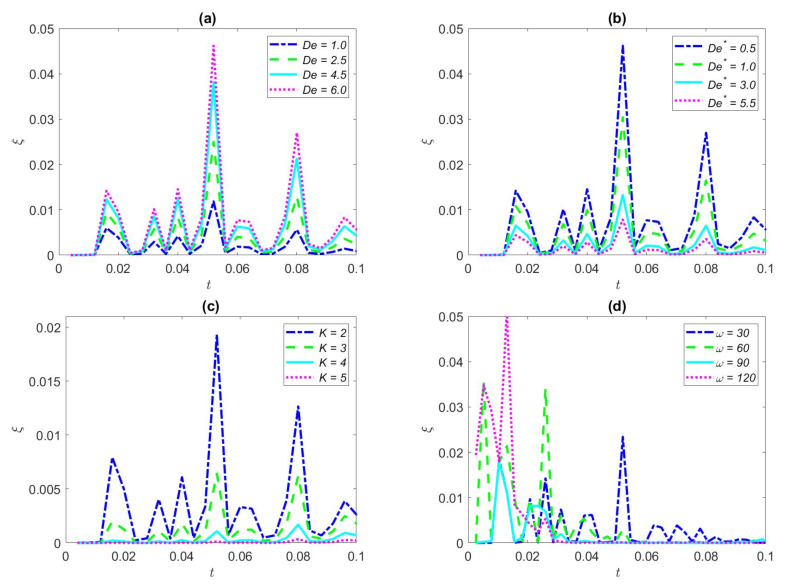
Distribution of the EKEC efficiency of the triangular waveform for some values of *De*, *De*^*^, *K*, *ω.* (**a**) *De*^*^ = 0.5, *φ* = 2%, *ω* = 20, *Ha* = 2, *K* = 1; (**b**) *De* = *6*, *φ* = 2%, *ω* = 20, *Ha* = 2, *K* = 1; (**c**) *De* = 1, *De*^*^ = 0.5, *φ* = 2%, *ω* = 20, *Ha* = 1; (**d**) *De* = 1, *De*^*^ = 0.5, *φ* = 2%, *Ha* = 1, *K* = 1.

**Figure 9 nanomaterials-12-03355-f009:**
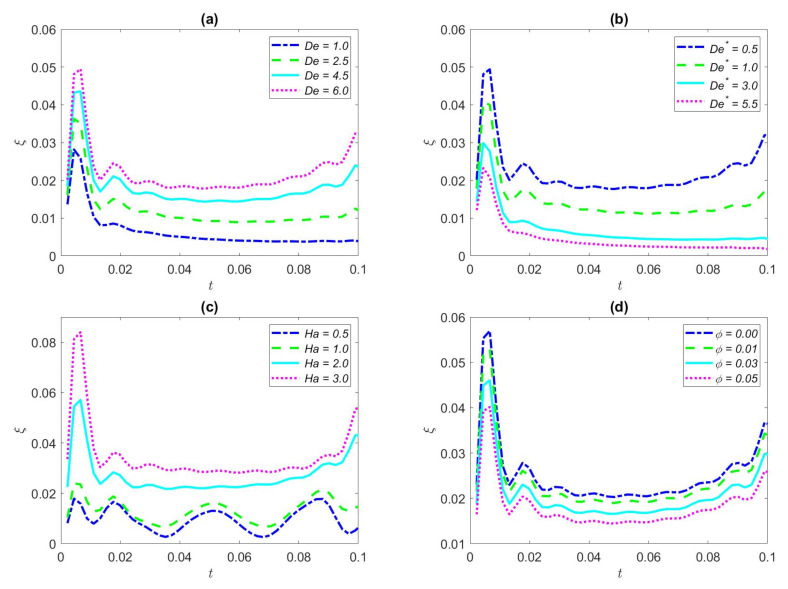
Distribution of the EKEC efficiency of the square waveform for some values of *De*, *De*^*^, *Ha*, *φ.* (**a**) *De*^*^ = 0.5, *φ* = 2%, *ω* = 2, *Ha* = 2, *K* = 1; (**b**) *De* = 6, *φ* = 2%, *ω* = 2, *Ha* = 2, *K* = 1; (**c**) *De* = 8, *De*^*^ = 0.5, *φ* = 2%, *ω* = 2, *K* = 1; (**d**) *De* = 6, *De*^*^ = 0.5, *ω* = 2, *Ha* = 2, *K* = 1.

**Figure 10 nanomaterials-12-03355-f010:**
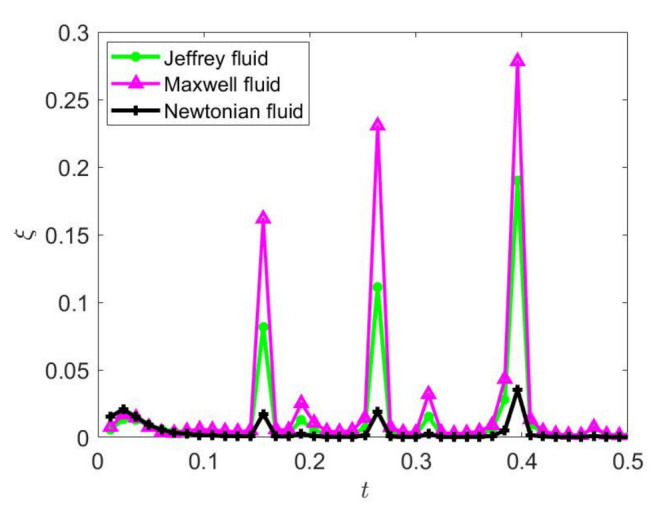
Comparison of EKEC efficiencies with dimensionless time variation for three fluids in the condition of square waves at *φ* = 2%, *ω* = 2, *Ha* = 2, *K* = 1.

## Data Availability

Not applicable.

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
