# Peer review of "The Impacts of Viscoelastic Behavior on Electrokinetic Energy Conversion for Jeffreys Fluid in Microtubes"

_nanomaterials, 2022, doi:10.3390/nano12193355_

Round 1
Reviewer 1 Report
In the paper under review, the authors study the electro-kinetic energy conversion efficiency, streaming potential of viscoelastic fluids in a microtube subject to an external axial pressure gradient and a transverse magnetic field. By the Green function method, the authors obtain the analytical solution for the axial velocity. The influence of different parameters on the model is analyzed and illustrated by various figures.
Undoubtedly, the problem under consideration is interesting from both theoretical and applied points of view. However, I have concerns about the rheological model (5) used by the authors.
1) The authors use the Jeffrey model (5) and give numerous references, but no pioneering work among those references. The Jeffrey model cannot be found in classical monographs and is not universally accepted. On the other hand, the model Jeffreys model (in other terminology, the Oldroyd model )is well-known, see, for example,
Saut, J.-C. Lectures on the mathematical theory of viscoelastic fluids. In: Lect. Anal. Nonlinear Partial Differ. Equ. Series Morningside Lect. Math. Series, pp. 325–393 (2013)
Who first proposed the Jeffrey model? Are Jeffrey and Jeffreys different people?
2) Moreover, in the rheological relation, the authors use the partial time-derivative, although in modern mathematical modeling it is customary to use the Jaumann objective derivative:
E. S. Baranovskii, Steady flows of an Oldroyd fluid with threshold slip, Commun. Pure Appl. Anal., 18:2 (2019), 735-750
https://doi.org/10.3934/cpaa.2019036
or the Oldroyd objective derivative:
C. Le Roux, On flows of viscoelastic fluids of Oldroyd type with wall slip, J. Math. Fluid Mech., 16 (2014), 335-350, https://doi.org/10.1007/s00021-013-0159-9
This is proposed to ensure that flow models satisfy the principle of material objectivity, that is, the form of the corresponding constitutive equation does not change after a change of variables. Does equation (5) satisfy the principle of material objectivity?
I ask the authors to address these comments carefully in order to provide more valuable work for the scientific community.
Author Response
1) The authors use the Jeffrey model (5) and give numerous references, but no pioneering work among those references. The Jeffrey model cannot be found in classical monographs and is not universally accepted. On the other hand, the model Jeffreys model (in other terminology, the Oldroyd model )is well-known, see, for example,
Saut, J.-C. Lectures on the mathematical theory of viscoelastic fluids. In: Lect. Anal. Nonlinear Partial Differ. Equ. Series Morningside Lect. Math. Series, pp. 325–393 (2013)
Who first proposed the Jeffrey model? Are Jeffrey and Jeffreys different people?
Response:Thanks for the careful discovery. Due to carelessness, we are sorry to make such a mistake.The rheological model (5) refers to the following literature.
Mederos, G.; Arcos, J.; Bautista, O.; Méndez, F. Hydrodynamics rheological impact of an oscillatory electroosmotic flow on a mass transfer process in a microcapillary with a reversible wall reaction. Phys. Fluids, 2020, 32:122003.
Koner, P.; Bera, S.; Ohshima, H. Effect of ion partitioning on an oscillatory electro-osmotic flow on solute transport process of fractional Jeffrey fluid through polyelectrolyte-coated nanopore with reversible wall reaction. Phys. Fluids,2022, 34: 062016.
Byron, B.R.; Robert, C.A.; Ole H. The General Linear Viscoelastic Fluid. In Dynamics of Polymeric Liquids; A Wiley-Interscience Publication, Canada,1987; Volume 1, pp.255-283.
The references were further supplemented.And Jeffrey and Jeffreys are different, we have changed Jeffrey to Jefferys in the article.The introduction is also augmented. These will be reflected in the updated article.
2) Moreover, in the rheological relation, the authors use the partial time-derivative, although in modern mathematical modeling it is customary to use the Jaumann objective derivative:
E. S. Baranovskii, Steady flows of an Oldroyd fluid with threshold slip, Commun. Pure Appl. Anal., 18:2 (2019), 735-750
https://doi.org/10.3934/cpaa.2019036
or the Oldroyd objective derivative:
C. Le Roux, On flows of viscoelastic fluids of Oldroyd type with wall slip, J. Math. Fluid Mech., 16 (2014), 335-350, https://doi.org/10.1007/s00021-013-0159-9
This is proposed to ensure that flow models satisfy the principle of material objectivity, that is, the form of the corresponding constitutive equation does not change after a change of variables. Does equation (5) satisfy the principle of material objectivity?
Response:Thanks to the reviewer for a very insightful question, we will make the following explanation.
First, let's introduce the general linear Jeffreys model. We can include the time derivative of the rate-of-strain tensor γand get the constitutive equation
(1)
which is known as the Jeffreys model, τ is the stress tensor, η0 is the zero-shear-rate viscosity. This equation, containing two time constants λ1, and λ2 (the "relaxation time" and the "retardation time", respectively).
Because the Jeffreys model of Eq. (1) is known to describe linear viscoelastic behavior qualitatively, we use it to generate a quasilinear model. This is done by replacing the partial time derivatives in the differential form of the Jeffreys model with the convected time derivatives to obtain the convected Jeffreys model' or Oldroyd's fluid B:
(2)
is the firs trate-of-strain tensors, is the second rate-of-strain tensors. The convectedJeffreys model contains several other models as special cases:
- If λ2= 0, the model reduces to the "convected Maxwell model." The convected Maxwell model has been widely used for viscoelastic flow calculations, because of its simplicity.
- Ifλ1= 0, the model simplifies to a second-order fluid with a vanishing second normal stress coefficient.
- If λ1= λ2, the model reduces to a Newtonian fluid with viscosity η0.
The above model refers to the following book.
Byron, B.R.; Robert, C.A.; Ole H. The General Linear Viscoelastic Fluid. In Dynamics of Polymeric Liquids; A Wiley-Interscience Publication, Canada,1987; Volume 1, pp.255-283.
The study of the linear Jeffreys model is shown in the following articles:
Mederos, G.; Arcos, J.; Bautista, O.; Méndez, F. Hydrodynamics rheological impact of an oscillatory electroosmotic flow on a mass transfer process in a microcapillary with a reversible wall reaction. Phys. Fluids, 2020, 32:122003.
Koner, P.; Bera, S.; Ohshima, H. Effect of ion partitioning on an oscillatory electro-osmotic flow on solute transport process of fractional Jeffrey fluid through polyelectrolyte-coated nanopore with reversible wall reaction. Phys. Fluids,2022, 34: 062016.
Linear model is simple to solve and deal with the problem.The convected Jeffreys model mentioned by the reviewer is not easy to solve for the specific dimensionless velocity, flow potential and electric energy conversion efficiency. The current range of capabilities is not sufficiently to handle the quasi-linear differential models. In the subsequent research, we will study and explore more in-depth work.

Reviewer 2 Report
Review of the manuscript The impacts of viscoelastic behavior on electrokinetic energy conversion for Jeffrey fluid in microtubes by Li et al.
In this work, the electrokinetic energy conversion efficiency is analyzed, considering the fluid flow is driven by the simultaneous effects of oscillatory pressure and electric and magnetic forces. The fluid viscoelastic behavior follows the Jeffreys model. The governing equations are solved for the unidirectional case, which allows for simplification of the mathematical model. The analytical solution is derived based on the Green function method. The influence of the dimensionless parameters in the analysis on the hydrodynamic field and the streaming potential is analyzed.
I have revised the manuscript, and I ask the authors to address the following points before making a recommendation
- The Abstract should mention that the two Deborah numbers are associated with the Jeffreys fluid's relaxation and retardation time.
- In the Abstract, the main findings of this work must be mentioned in brevity.
- Page 1, line 27: "Stern" must be written instead of "stern"
- Line 28: In some cases, the ion distribution does not follow the Boltzmann distribution. Please, verify the text.
- In the Introduction section, for instance, in lines 34 and 35 of page 1 (or in lines 70-81 on page 2), the following paper should be mentioned: https://doi.org/10.1063/5.0027818
- In the Introduction section, the novelty of this work compared to the published literature must be emphasized.
- Also, what is the motivation to include an oscillatory pressure gradient in this flow analysis? From a physical point of view, what are the consequences of such a consideration? What is the hypothesis for considering a nanofluid, the viscoelasticity, and the magnetic field in the analysis? The above points must be addressed.
- The title of Section 2. Materials and Methods should be changed, maybe as a Set up of the problem or any other description that corresponds to the mathematical formulation of the studied phenomenon.
- page 3, line 101: if there is an oscillatory pressure gradient, \Delta p can be positive or negative and not only > 0.
- In Fig. 1, the oscillatory character of the pressure gradient must be shown.
- Page 3, line 122, What is the reason for mentioning the Resonance behavior in this part of the manuscript?
- The section describing the electric field should be shortened. Such a theory is widely known, Eqs (6)-(14).
- In Eq. (15), I do not understand the scaling of \bar \omega. Please explain it. Why do you scale \Omega concerning the diffusion time? It has not to make sense.
- page 5, line 156: for $\lamda_2$, it should be "retardation time" instead of "relaxation time"
- page 6, line 162: Explain a physical point of view of the dimensionless parameters arising in the analysis
- It is imperative to include a validation of the results derived in this work and compare it against other works published in the past in some appropriate limits.
- page 9, does \eta_0 correspond to that of a viscoelastic fluid? In addition, this physical parameter should be affected by the nanoparticles. Some references must be included to verify the values of the physical parameters.
- Line 218-243, page 9: Could you show that the combination of the physical parameters provides the values of the dimensionless parameters used for the numerical calculations?
- When describing the figures, in several cases, only trends are mentioned; however, the explanation of the results must be based on fundamental aspects of the transport phenomena involved in the analysis.
- In Fig. 7 (c), a value 0f K=0.5, 1,1.5 are used. However, under these circumstances, the EDL is overlapped. The above would modify the solution for the induced electric field in the EDL, and the solution given by Eq. (14) is no longer valid. Accordingly, verify those calculations made with these values in the whole manuscript.
- Another point, Why is the time set up to t=0.1 in figures 5-8?
Author Response
Review of the manuscript The impacts of viscoelastic behavior on electrokinetic energy conversion for Jeffrey fluid in microtubes by Li et al.
In this work, the electrokinetic energy conversion efficiency is analyzed, considering the fluid flow is driven by the simultaneous effects of oscillatory pressure and electric and magnetic forces. The fluid viscoelastic behavior follows the Jeffreys model. The governing equations are solved for the unidirectional case, which allows for simplification of the mathematical model. The analytical solution is derived based on the Green function method. The influence of the dimensionless parameters in the analysis on the hydrodynamic field and the streaming potential is analyzed.
I have revised the manuscript, and I ask the authors to address the following points before making a recommendation
- The Abstract should mention that the two Deborah numbers are associated with the Jeffreys fluid's relaxation and retardation time.
Response:According to the reviewer's suggestion, we have made the change. These will be reflected in the updated article.
- In the Abstract, the main findings of this work must be mentioned in brevity.
Response: According to the reviewer's suggestion, we have made the change. These will be reflected in the revised article.
- Page 1, line 27: "Stern" must be written instead of "stern"
Response:Thanks for the careful criticism and correction.We have changed " stern " to " Stern ".
- Line 28: In some cases, the ion distribution does not follow the Boltzmann distribution. Please, verify the text.
Response: Thanks for the careful criticism and correction.We added in the text that the ions follow a Boltzmann distribution in a quasi-steady-state environment.
- In the Introduction section, for instance, in lines 34 and 35 of page 1 (or in lines 70-81 on page 2), the following paper should be mentioned: https://doi.org/10.1063/5.0027818
Response:Thanks for the suggestion. We have put the above literature in the introduction section. These will be reflected in the revised article.
- In the Introduction section, the novelty of this work compared to the published literature must be emphasized.
Response:Thanks for the patient guidance. According to the reviewer’s request, we have carefully corrected it, which is included in our manuscript.
- Also, what is the motivation to include an oscillatory pressure gradient in this flow analysis? From a physical point of view, what are the consequences of such a consideration? What is the hypothesis for considering a nanofluid, the viscoelasticity, and the magnetic field in the analysis? The above points must be addressed.
Response:According to the reviewer's suggestion, we will make the following explanation.
This is an oscillating problem, and oscillating pressure gradients can generate periodic pressures, which further generate the periodic streaming potentials, resulting in time-evolving electrokinetic energy conversion efficiency. We set up a nanofluid model with Jeffreys fluid constitutive relation, and study the electrokinetic energy conversion efficiency problem under the coupling action of magnetic field and periodic pressure gradient.
- The title of Section 2. Materials and Methods should be changed, maybe as a Set up of the problem or any other description that corresponds to the mathematical formulation of the studied phenomenon.
Response:Thanks for the patient guidance. According to the reviewer’s request,we have carefully corrected the title of Section 2 to Problem formulation.
- page 3, line 101: if there is an oscillatory pressure gradient, \Delta p can be positive or negative and not only > 0.
Response:Sorry to make such mistakes.We have corrected itin the article.
- In Fig. 1, the oscillatory character of the pressure gradient must be shown.
Response:According to the suggestion of the reviewer, we have corrected it. See Fig. 1 for details.
- Page 3, line 122, What is the reason for mentioning the Resonance behavior in this part of the manuscript?
Response:According to reviewer’s question, we will make the following explanation.
This is due to the elastic effect of the fluid, periodic pressure causes periodic electric field, this behavior is called resonance [51].
- The section describing the electric field should be shortened. Such a theory is widely known, Eqs (6)-(14).
Response:Thanks for suggestion. According to reviewer’s suggestion,we have shortened the description of the electric field. These will be reflected in the revised article.
- In Eq. (15), I do not understand the scaling of \bar \omega. Please explain it. Why do you scale \Omega concerning the diffusion time? It has not to make sense.
Response:According to the reviewer's suggestion, we will make the following elaboration.
ω is the introduced dimensionless frequency to facilitate subsequent derivation and calculation.
- page 5, line 156: for $\lamda_2$, it should be "retardation time" instead of "relaxation time"
Response:Thanks for suggestion. We have modified λ2to be retardation time.
- page 6, line 162: Explain a physical point of view of the dimensionless parameters arising in the analysis
Response:Under the current International Vocabulary of Metrology, all counted quantities and all ratios of two quantities of the same kind are called quantities of dimension one, or alternately dimensionless quantities. The definition is ‘‘quantity for which all the exponents of the factors corresponding to the base quantities in its quantity dimension are zero.” The unit of measurement for such quantities is 1, the algebraic result of setting all of the exponents to zero. We have distinguished physical parameters from dimensionless quantities for easy understanding. Change to" , , , , , , , and ."
- It is imperative to include a validation of the results derived in this work and compare it against other works published in the past in some appropriate limits.
Response:According to the suggestion of the reviewer, we compare the streaming potential and EKEC efficiency between the present result and that of Gao et al. [44]. As shown in the figures:
Figure 2. The present results are compared with thestreaming potential (a) and EKEC efficiency (b) of Gao et al. [44] in (a) De=1, φ=2%, ω=60, Ha=2, K=2; (b) De=1, φ=2%, ω=2, Ha=1, K=1.5;
When the λ2 is zero, the Jeffreys model in this paper becomes the Maxwell model. Under the condition of keeping other parameters consistent, the streaming potential and EKEC efficiency are basically consistent with the research results of Gao et al. [44].
- page 9, does \eta_0 correspond to that of a viscoelastic fluid? In addition, this physical parameter should be affected by the nanoparticles. Some references must be included to verify the values of the physical parameters.
Response:Thanks for careful discovery. correspond to that of a viscoelastic nanofluid fluid, we add the sentences:η0=µf/(1-φ)2.5is the zero shear viscosity of the fluid; µfis the viscosity of the base fluid.µf is the viscosity of the base fluid. μf= 8.91 × 10-3 kg m-1s-1.According to the suggestion of the reviewer, we change the image in the revised article.
- Line 218-243, page 9: Could you show that the combination of the physical parameters provides the values of the dimensionless parameters used for the numerical calculations?
Response: According to the suggestion of the reviewer, we change the sentences. These will be reflected in the revised article.Under the current International Vocabulary of Metrology, all counted quantities and all ratios of two quantities of the same kind are called quantities of dimension one, or alternately dimensionless quantities. The definition is ‘‘quantity for which all the exponents of the factors corresponding to the base quantities in its quantity dimension are zero.” The unit of measurement for such quantities is 1, the algebraic result of setting all of the exponents to zero.The combination of the physical parameters provides the values of the dimensionless parameters used in MATLAB.
- When describing the figures, in several cases, only trends are mentioned; however, the explanation of the results must be based on fundamental aspects of the transport phenomena involved in the analysis.
Response:According to the suggestion of the reviewer, we change the sentences. These will be reflected in the revised article.
- In Fig. 7 (c), a value 0f K=0.5, 1,1.5 are used. However, under these circumstances, the EDL is overlapped. The above would modify the solution for the induced electric field in the EDL, and the solution given by Eq. (14) is no longer valid. Accordingly, verify those calculations made with these values in the whole manuscript.
Response: Thanks for careful discovery.In Fig. 7(c), we have modified K to be 2, 3, 4, 5. The corresponding figure and explanation section are as follows.
Figure 8(c) refers to the variation of the EKEC efficiency at various values of di-mensionless electro-kinetic widths (K =2, 3, 4, and 5, respectively) in the case of trian-gular waveform. From Figure 8(c), we can clearly see that a larger electric width leads to a smaller EKEC efficiency. As expected, as the dimensionless K expands, this means that the width of the EDL is shrinking, which leads to a decrease in the number of EDL ions, thus dropping the efficiency of the EKEC. EDL width is bitty, the greater the den-sity of free charged ions in microchannel, the greater the resistance of fluid movement, leading to fluid motion wave is more and more gradual. Consequently, the bigger K turns more smoothly to the fluctuation of EKEC efficiency.
- Another point, Why is the time set up to t=0.1 in figures 5-8?
Response:According to the reviewer's suggestion, we will make the following elaboration.
We set t=0.1 because the effect of the graph at this time is more obvious, which is convenient for explaining the problem. Make t larger as shown in the following figure:

Round 2
Reviewer 1 Report
The authors gave detailed answers to my questions and improved the quality of the manuscript. Some previously incomprehensible points have become clear, but new questions and concerns have arisen.Point 1. I agree that the linearization of the constitutive equation greatly simplifies the study of the flow model. However, such a simplification leads to the fact that many important nonlinear effects cannot be taken into account within the framework of the obtained model. Why is there a lack of modern computational capabilities to investigate the convective Jeffreys (Oldroyd-B) model? I think that the authors should take a closer look at the analysis of such possibilities and find a way out of the current situation.
Point 2. I think the equation (5) is wrong. I mean the minus sign, which is on the right side in front of the viscosity eta_0. The correct version of the Jeffreys constitutive equation can be found in
https://doi.org/10.1007/s10440-016-0076-z
See formula (2.1) on Page 198.
Point 3. In their response, the authors wrote: ''If λ_1= λ_2, the model reduces to a Newtonian fluid with viscosity η_0''. This statement is obviously false.
Point 4. I believe that the authors need to mention in the manuscript the results of previous works, in which other researchers considered the model of Jeffreys (Oldroyd-B) with an objective derivative (Jaumann derivative, upper convected derivative, lower convected derivative, etc.) instead of the time derivative. Otherwise, the literature review will be incomplete.
Point 5. On Figure 1, the terminology ''Jeffrey fluid'' remains and it is incorrect.
Author Response
The authors gave detailed answers to my questions and improved the quality of the manuscript. Some previously incomprehensible points have become clear, but new questions and concerns have arisen.
Point 1. I agree that the linearization of the constitutive equation greatly simplifies the study of the flow model. However, such a simplification leads to the fact that many important nonlinear effects cannot be taken into account within the framework of the obtained model. Why is there a lack of modern computational capabilities to investigate the convective Jeffreys (Oldroyd-B) model? I think that the authors should take a closer look at the analysis of such possibilities and find a way out of the current situation.
Response: Thanks to the reviewer for a very insightful question, we will make the following explanation.
We refer to the partial nonlinear model, which is similar with the nonlinear terms presented in references:
- Prasad,,V.,Ramachandra,Gaffar,,S.,Abdul,Reddy,,E.,Keshava,Beg,,O.,& Anwar.(2015).Numerical study of non-Newtonian Jeffreys fluid from a permeable horizontal isothermal cylinder in non-Darcy porous medium.JOURNAL OF THE BRAZILIAN SOCIETY OF MECHANICAL SCIENCES AND ENGINEERING,37(6),1765-1783.
- Krishna,,M.,& Veera.(2020).Hall and ion slip impacts on unsteady MHD free convective rotating flow of Jeffreys fluid with ramped wall temperature.INTERNATIONAL COMMUNICATIONS IN HEAT AND MASS TRANSFER,119.
- Prasad,,V.,Ramachandra,Gaffar,,S.,Abdul,Reddy,,E.,Keshava,Beg,,O.,& Anwar.(2014).Flow and Heat Transfer of Jeffreys Non-Newtonian Fluid from Horizontal Circular Cylinder.JOURNAL OF THERMOPHYSICS AND HEAT TRANSFER,28(4),764-770.
Generally, there are numerous methods for dealing with nonlinear terms, such as the regular and singular perturbation methods, similarity transformations, and linear transformations.Our current work is investigating the EKEC efficiency under periodic pressure gradients, considering three kinds of periodic waves, especially the complex superposition waves of square and triangular waves.If the problem of nonlinear terms is considered, it will considerably increase the difficulty of processing the problem, and it is not easy to solve the velocity, flow potential and EKEC efficiency by Green function method. In this study, we mainly try to develop the Green function method in EKEC efficiency of Jeffreys model in some simple case. In view of the above concerns, we adopt the linear Jeffreys model to deal with the EKEC efficiency problem, and we will study and explore more in-depth work in the subsequent research.
Point 2. I think the equation (5) is wrong. I mean the minus sign, which is on the right side in front of the viscosity eta_0. The correct version of the Jeffreys constitutive equation can be found in
https://doi.org/10.1007/s10440-016-0076-z
See formula (2.1) on Page 198.
Response:Thanks to the reviewer for a very insightful question. We've got rid of the minus sign. The minus sign in equation (5) and the minus sign in front of the of equation (1) are eliminated in the solution process. For ease of writing and better understanding, the equation (1) and equation (5) are as follows:
Point 3. In their response, the authors wrote: ''If λ_1= λ_2, the model reduces to a Newtonian fluid with viscosity η_0''. This statement is obviously false.
Response: Thanks for the careful discovery. Due to carelessness, we are sorry to make such a mistake. It should be changed to “If λ1= λ2=0, the model reduces to a Newtonian fluid with viscosityη0.”
Point 4. I believe that the authors need to mention in the manuscript the results of previous works, in which other researchers considered the model of Jeffreys (Oldroyd-B) with an objective derivative (Jaumann derivative, upper convected derivative, lower convected derivative, etc.) instead of the time derivative. Otherwise, the literature review will be incomplete.
Response: According to the reviewer's suggestion, we have made the change. The modification of the introduction section and the added references are as follows.
The Jeffreys model (in other terminology, the Oldroyd-B model) is well known. In previous studies, numerous researchers have studied the fluid flow problem in steady state. The Jeffreys (Oldroyd-B) model with objective derivative has been studied by various scholars, such as Jaumann derivative, upper convective derivative, lower convective derivative, etc. [39-43]. A three-dimensional steady flow mathematical model of an incompressible Oldroyd-type viscoelastic fluid in a bounded region was studied by Evgenii [44]. The application of the original method to the weak solvability of the initial boundary value problem of Oldroyd model with regularized target Jaumann derivative and the study of attractors was proved by Zvyagin et al. [45]. Angiolo et al. [46]studied three dimensional nonlinear viscoelastic models including stress relaxation and creep/recovery phenomena. Kaushik et al. [47] combinedthe elasticity and viscosity of viscoelastic fluids to study the initiated electroosmotic flow of viscoelastic fluids in a rotating rectangular microfluidic channel described by the Oldroyd-B model. However, the linear Jefferys model with respect to time derivatives has also yielded countless results [48-50]. Mederos et al. [51] performed a theoretical analysis of the mass transport rate of lengthy microcapillaries with reactive walls connecting two reservoirs with different concentrations of electrically neutral solutes caused by oscillating electroosmotic flow of Jeffreys fluids. Koner et al. [52] investigated the effect of ion partitioning on solute transport phenomena in fractional Jeffreys fluids. Liu et al. [53] studied the unsteady electroosmotic flow of linear Jeffreys fluids in microbial pipes. The Jeffreys fluid model for studying the rheology of fluids with ciliary motion was investigated by Ishtiaqet al. [54]. Ambreen et al. [55] focused on the heat transfer properties in oblique stagnation flow of Jeffreys fluids deformed by linear stretchable surfaces.
Finally, to the best of authors’ knowledge,relativelyminor EKEC efficiency research has been carried out on viscoelastic fluids, particularly on nanofluids with viscoelastic behaviour. Accordingly, this article investigates the influences of periodic pressure gradients and magnetic fields with three distinct waveforms (cosine, triangle, square) on the flow potential, axial velocity and EKEC efficiency of viscoelastic nanofluid. This is an oscillating problem, and oscillating pressure gradients can generate periodic pressures, which additionally generate periodic streaming potentials, resulting in time-evolving EKEC efficiency. Gao et al. [56] studied the simple Maxwell model, in this paper, instead of using convective derivatives like the Oldroyd-B model, we investigate the Jeffreys model using time derivatives as a simpler linear model. The Jeffreys fluid model is a straightforward viscoelastic fluid model, and it is of great significance to study the viscoelasticity of polymer fluids. We consider non-steady state and related dimensionless parameters and visualize the impact of each dimensionless parameter on the fluid flow behavior within the microtubes. Especially, the analytic solution for axial velocity field is computed obtaining the Green's function method. It is hoped these findings are able to offer relevant suggestions to the studies in Microfluidics and provide some reference points for future research.
- Christiaan, L.R. On Flows of Viscoelastic Fluids of Oldroyd Type with Wall Slip. J. Math. Fluid Mech., 2014, 16(2),335-350.
- Baranovskii, E.S. On steady motion of viscoelastic fluid of Oldroyd type. Sb. Math. 2014, 205(6),763-776.
- Antony N.B. Continuum mechanics modeling of complex fluid systems following Oldroyd's seminal 1950 work. J. Non-Newton. Fluid 2021, 298:104677.
- Baranovskii, E.S.; Artemov, M.A. Global Existence Results for Oldroyd Fluids with Wall Slip. Acta Appl. Math., 2017, 147(1),197-210.
- Hinch, J.; Harlen, O. Oldroyd B, and not A? J. Non-Newton. Fluid 2021 298:104668.
- Evgenii, S.B. STEADY FLOWS OF AN OLDROYD FLUID WITH THRESHOLD SLIP. Acta Metall. Sin., 2019,18(2),735-750.
- Zvyagin, V.G.; Vorotnikov, D.A. Approximating-topological methods in some problems of hydrodynamics. J. Fix Point Theory A.,2008, 3(1),23-49.
- Angiolo, F.; Lorenzo, F.; Fabio R.; Giuseppe S. Creep, recovery and vibration of an incompressible viscoelastic material of the rate type: Simple tension case. Int. J. Nonlin. Mech. 2022, 138:103851.
- Kaushik, P.; Abhimanyu, P.; Mondal, P.K.; Chakraborty; S. Confinement effects on the rotational microflows of a viscoelastic fluid under electrical double layer phenomenon. J. Non-Newton. Fluid 2017,244:123-137.
- Alvi, N.; Latif, T.; Hussain, Q.; Asghar, S. Peristalsis of nonconstant viscosity Jeffrey fluid with nanoparticles. Results phys. 2016,6:1109-1125.
- Mederos, G.; Arcos, J.; Bautista, O.; Méndez, F. Hydrodynamics rheological impact of an oscillatory electroosmotic flow on a mass transfer process in a microcapillary with a reversible wall reaction. Phys. Fluids, 2020, 32:122003.
- Koner, P.; Bera, S.; Ohshima, H. Effect of ion partitioning on an oscillatory electro-osmotic flow on solute transport process of fractional Jeffrey fluid through polyelectrolyte-coated nanopore with reversible wall reaction. Phys. Fluids,2022, 34: 062016.
- Liu, Q.S.; Yang, L.G.; Su, J. Transient electroosmotic flow of general Jeffrey fluid between two micro-parallel plates. Acta Phys. Sin. 2013,14:144702.
- Ishtiaq, F.; Ellahi, R.; Bhatti, M.M.; Alamri, S.Z. Insight in Thermally Radiative Cilia-Driven Flow of Electrically Conducting Non-Newtonian Jeffrey Fluid under the Influence of Induced Magnetic Field. Math. 2022,10(12).
- Ambreen, A.; Sajid, M.; Rana, M. A.; Mahmood, K. Numerical simulation of heat transfer features in oblique stagnation-point flow of Jeffrey fluid. Ambreen. Aip Adv. 2018,8(10).
- Gao, X.; Zhao, G.P.; Li, N.; Zhang, J.L.; Jian, Y.J. The electrokinetic energy conversion analysis of viscoelastic fluid under the periodic pressure in microtubes. Colloid Surf. A, 2022, 646:128976.
Point 5. On Figure 1, the terminology ''Jeffrey fluid'' remains and it is incorrect.
Response: According to the suggestion of the reviewer, we have corrected it. See Fig. 1 for details.
Figure 1. depicts a sort of EKEC system.

Reviewer 2 Report
The authors have addressed my comments from the first review. Therefore, I suggest accepting the manuscript to be published inNanomaterials.
Round 3
Reviewer 1 Report
The authors gave detailed answers to my questions and made the corresponding corrections to their manuscript. Initially unclear points received appropriate explanations. I agree with the authors answers.
Conclusion: I recommend the paper for publication in Nanomaterials. However, before publication, the following corrections should be made due to some typographical errors and inaccuracies:
1) Line 114: relevantsuggestions --> relevant suggestions.
2) Line 184: followinginitial --> following initial.
3) Line 229: derivedfrom --> derived from.
4) Line 611: Evgenii, S.B. --> Baranovskii, E.S.
5) Line 611: Acta Metall. Sin --> Commun. Pure Appl. Anal.
6) Line 631: ElectricallyConducting --> Electrically Conducting.
7) The reference [45] (Zvyagin and Vorotnikov) is not relevant and should be removed from the references list.
Author Response
The authors gave detailed answers to my questions and made the corresponding corrections to their manuscript. Initially unclear points received appropriate explanations. I agree with the authors answers.
Conclusion: I recommend the paper for publication in Nanomaterials. However, before publication, the following corrections should be made due to some typographical errors and inaccuracies:
1) Line 114: relevantsuggestions --> relevant suggestions.
2) Line 184: followinginitial --> following initial.
3) Line 229: derivedfrom --> derived from.
4) Line 611: Evgenii, S.B. --> Baranovskii, E.S.
5) Line 611: Acta Metall. Sin --> Commun. Pure Appl. Anal.
6) Line 631: ElectricallyConducting --> Electrically Conducting.
7) The reference [45] (Zvyagin and Vorotnikov) is not relevant and should be removed from the references list.
Response: Thanks for the patient guidance. According to the reviewer’s request, we have carefully corrected the above error, and reference [45] has been deleted.
